# Prot2Text: Multimodal Protein's Function Generation with GNNs and Transformers

**Hadi Abdine**
LIX, École Polytechnique, IP Paris
hadi.abdine@polytechnique.edu
Palaiseau, France

**Michail Chatzianastasis**
LIX, École Polytechnique, IP Paris
Palaiseau, France
mixalisx97@gmail.com

**Costas Bouyioukos**
Epigenetics and Cell Fate, CNRS UMR7216
Université Paris Cité
Paris, France
costas.bouyioukos@u-paris.fr

**Michalis Vazirgiannis**
LIX, École Polytechnique, IP Paris
Palaiseau, France
mvazirg@lix.polytechnique.fr

## Abstract

In recent years, significant progress has been made in the field of protein function prediction with the development of various machine-learning approaches. However, most existing methods formulate the task as a multi-classification problem, i.e. assigning predefined labels to proteins. In this work, we propose a novel approach, **Prot2Text**, which predicts a protein's function in a free text style, moving beyond the conventional binary or categorical classifications. By combining Graph Neural Networks(GNNs) and Pretrained Language Models, in an encoder-decoder framework, our model effectively integrates diverse data types including protein sequence, structure, and textual annotation and description. This multimodal approach allows for a holistic representation of proteins' functions, enabling the generation of detailed and accurate functional descriptions. To evaluate our model, we extracted a multimodal protein dataset from SwissProt, and demonstrate empirically the effectiveness of Prot2Text. These results highlight the transformative impact of multimodal models, specifically the fusion of GNNs and pretrained transformer-based language models, empowering researchers with powerful tools for more accurate function prediction of existing as well as first-to-see proteins.

## 1 Introduction

Understanding proteins' function is a central problem in biological sciences, as proteins are the fundamental elements of almost all biological functions. Accurate prediction of proteins' function is essential for understanding biological systems as well as for various applications, such as drug discovery, enabling researchers to identify and target specific proteins that play critical roles in disease pathways (Ha *et al.*, 2021). Traditionally, proteins' functions prediction has been approached through classification methods, assigning predefined labels to proteins based on their characteristics (Kulmanov and Hoehndorf, 2019). However, this approach often oversimplifies the complexity of proteins' functionality, limiting the depth of our understanding. To overcome this limitation, we propose a novel view on proteins' functions prediction based on reformulating the task using free-text proteins' descriptions instead of relying on predefined labels. The rapid progress in transformer-based models has brought a massive revolution to the field of Natural Language Processing (NLP). These models have demonstrated impressive language generation capabilities, allowing them to perform a wide range of NLP tasks with remarkable performance, including text completion, translation, sentiment

NeurIPS 2023 AI for Science Workshop.

analysis and question-answering (Vaswani *et al.*, 2017; Radford *et al.*, 2019; Brown *et al.*, 2020). On the other hand, Graph Neural Networks(GNNs) have emerged as a powerful tool for modeling graph-structured data, capturing the intricate relationships between different elements in a graph (Kipf and Welling, 2017; Reiser *et al.*, 2022). However, the integration of GNNs and transformers faces various challenges, such as effectively handling the heterogeneity of data representations, therefore the field is still in its early stages. Despite this, the potential benefits of leveraging both GNNs and transformers for graph-to-text applications, such as predicting the functional properties of proteins are substantial. To that end, we develop a novel multimodal framework, **Prot2Text**, that can generate detailed and accurate descriptions of proteins' functions in free text. We effectively integrate GNNs and Large Language Models (LLMs), to encompass both structural and sequential information of the protein's 3D structure and amino acid's sequence respectively. The encoder-decoder architecture forms the backbone of our model, with the encoder component employing a Relational Graph Convolution Network (RGCN) (Schlichtkrull *et al.*, 2018) to process the proteins' graphs and the ESM protein language model (Lin *et al.*, 2023a) to encode the protein's sequence. The decoder component utilizes a pre-trained GPT2 model to generate detailed proteins' descriptions. To train our multimodal model, we compile a dataset of proteins extracted from SwissProt, a comprehensive collection of protein annotations obtained from the UniProt database (Consortium, 2015). This dataset encompasses a vast number of proteins, each annotated with its corresponding function or description. In addition to the textual information, we obtain the 3D structure representation of the proteins from AlphaFold (Varadi *et al.*, 2022). We further release this curated dataset to the public, allowing other researchers to use it for benchmarking and further advancements in the field. Our main contributions can be summarized as follows:

- We introduce the **Prot2Text** framework, a novel multimodal approach for generating proteins' functions in free text. Our model combines both GNNs and ESM to encode the protein in a fused representation while a pre-trained GPT2 decodes the protein's text description.

- We propose various baselines for protein text generation and demonstrate that the integration of both graph and sequence protein information leads to better generation capabilities.

- We further release a comprehensive multimodal protein dataset, which includes $256,690$ protein structures, sequences, and textual function descriptions. Researchers can leverage this dataset to benchmark and compare their models, thereby driving advancements in the field and enabling for a more robust and standardized evaluation of proteins' functions prediction methods in free text format.

## 2   Related Work

**Transformers.**   The transformer-based encoder-decoder model was first introduced by Vaswani *et al.* (2017) in their paper "Attention is all you need". Since then, this model architecture has become the de-facto standard encoder-decoder architecture in Natural Language Processing (NLP). Despite going through significant research on different pre-training objectives for transformer-based encoder-decoder models such as T5 (Raffel *et al.*, 2019) and Bart (Lewis *et al.*, 2020), the model architecture has remained largely the same. Radford *et al.* took advantage of the transformer architecture (Vaswani *et al.*, 2017), which is superior and conceptually simpler than Recurrent Neural Networks to introduce the OpenAI GPT model. Specifically, they pre-trained a left-to-right transformer decoder as a general language model using the GPT architecture. Following, they fine-tuned the model on 12 different language understanding tasks by applying various transformations to the input. Later on, GPT-2 (Radford *et al.*, 2019) was introduced, a more advanced version of GPT having more trainable parameters. The authors showed that as long as general language models have very high capacities, they can reach reasonable performance on many specific natural language processing tasks. The use of the transformer's architecture is expanded later to include modalities other than natural language such as images (Dosovitskiy *et al.*, 2021), protein amino-acid sequences (Rives *et al.*, 2021; Lin *et al.*, 2023a), and molecules SMILES string (Fabian *et al.*, 2020; Chithrananda *et al.*, 2020). All the models above are pretrained with the Masked Language Modeling task (MLM) introduced in BERT (Devlin *et al.*, 2019) and are able mostly to perform discriminative tasks.

**Multimodal models.**   The success of the transformer's uni-modality tasks made this architecture broadly studied for multimodal representation learning. One example is The CLIP (Contrastive Language-Image Pre-training) model (Radford *et al.*, 2021) which is a transformer model that

facilitates cross-modal understanding between images and text. It combines a ViT vision encoder, with a transformer-based language encoder to learn joint representations of images and their associated textual descriptions. By leveraging transformers in both the vision and text encoders, the CLIP model benefits from their ability to capture long-range dependencies. Another example is the MolT5 (Edwards *et al.*, 2022) which is a self-supervised learning framework based on the T5 model (Raffel *et al.*, 2019) for pretraining models on a vast amount of unlabeled natural language text and molecule SMILES strings. MolT5 is able to perform bidirectional translation between molecule representations and natural language allowing molecule captioning and generation providing text prompts.

**Graph Neural Networks.** Graph neural networks (GNNs) have emerged as a powerful framework for modeling and analyzing graph-structured data (Scarselli *et al.*, 2009; Kipf and Welling, 2017). By iteratively exchanging and integrating information among nodes, GNNs can propagate and refine features throughout the graph, ultimately encoding a comprehensive understanding of the graph's structure and semantics. This ability to capture complex relationships within graphs has contributed to the success of GNNs in various domains, including social network analysis, recommendation systems, and bioinformatics (Zitnik *et al.*, 2018; Zhang *et al.*, 2021; Chatzianastasis *et al.*, 2023a). Numerous studies have suggested various enhancements and expansions to the GNNs' models. Some notable contributions include the introduction of more expressive and adaptable aggregation functions, such as those proposed by Murphy *et al.* (2019), Seo *et al.* (2019) and Chatzianastasis *et al.* (2023b). Moreover, several schemes have been developed to incorporate different local structures or high-order neighborhoods, as explored by Morris *et al.* (2020) and Nikolentzos *et al.* (2020). Furthermore, the domain of GNNs has expanded to encompass heterogeneous graphs, where nodes and edges can have different types and semantics, leading to the development of Heterogeneous Graph Neural Networks effectively handling such complex graph structures (Schlichtkrull *et al.*, 2018; Zhang *et al.*, 2019).

**Protein Representation Learning.** In the field of protein representation learning, various approaches have emerged over the years, aiming to capture meaningful information from proteins using different data modalities and computational techniques. One prominent avenue of research is focused on sequence-based representations, that extract features solely from the amino-acid sequences of proteins. To achieve this, deep learning techniques such as Recurrent Neural Networks (RNNs) and Convolutional Neural Networks (CNNs) have been applied, enabling the direct learning of representations from protein sequences (Liu, 2017; Bileschi *et al.*, 2019). Drawing inspiration from the remarkable achievements of language models in Natural Language Processing (NLP), researchers have also developed pre-trained language models tailored specifically for proteins (Brandes *et al.*, 2022; Lin *et al.*, 2023b). These models leverage large-scale protein datasets to learn powerful representations that can subsequently be utilized for various prediction tasks. In addition to sequence-based approaches, graph-based representations leverage the three-dimensional (3D) structure of proteins to capture their functional properties. Zhang *et al.* (2022) proposed a graph neural network model with a contrastive pertaining strategy for function prediction and fold classification tasks. Chen *et al.* (2023) proposed a 3D-equivariant graph neural network, specifically designed to estimate the accuracy of protein structural models. Wang *et al.* (2022) used a hierarchical graph network, which captures the hierarchical relations present in proteins and learns representations at different levels of granularity. Hybrid approaches integrate multiple data modalities, such as protein sequences, structures, and functional annotations, to create comprehensive representations. These methods combine the strengths of sequence-based and graph-based approaches to capture diverse aspects of protein function. Gligorijević *et al.* (2021) proposed DeepFRI which combines sequence features extracted from a pre-trained protein language model with protein structures. Our work aims to leverage protein sequence and structure models to generate free text annotations of proteins.

## 3   Methodology

In this section, we present our proposed multimodal framework, **Prot2Text**, for generating protein function descriptions in free text. A visual representation of the framework is provided in Figure 1.

**Graph Construction.** Upon obtaining the 3D proteins' structures using AlphaFold, we proceed to represent the proteins as a heterogeneous graph $G = (V, E, R)$, where $V = [N] := \{1, ..., N\}$ is the set of vertices representing the amino-acids of the proteins, $E \subseteq V \times V$ is the set of edges representing various interactions between the nodes and $R$ is a set of different edge interactions. Each

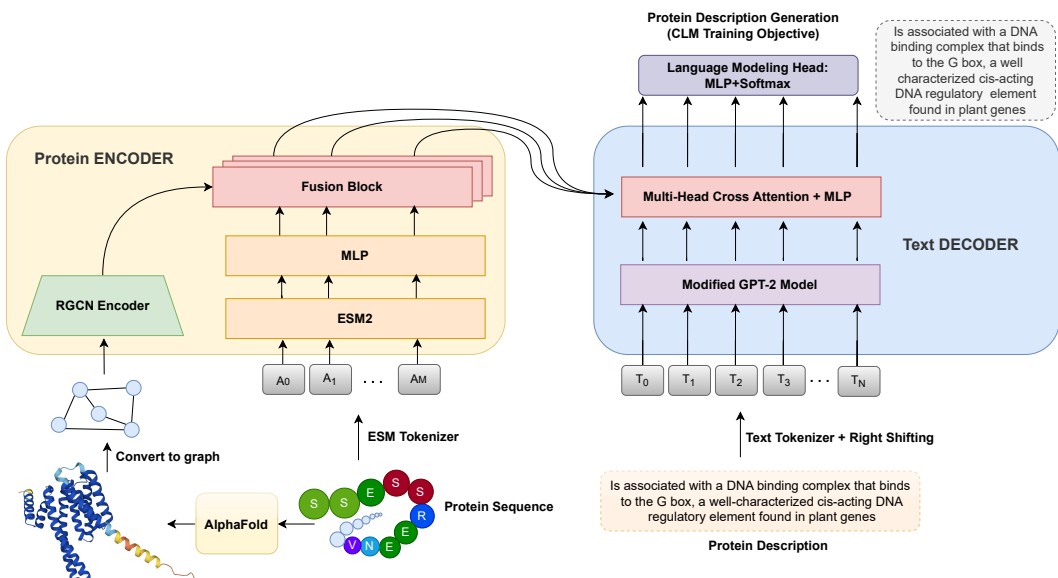

Figure 1: **Architecture of the proposed Prot2Text framework for predicting protein function descriptions in free text.** The model leverages a multimodal approach that integrates protein sequence, structure, and textual annotations. The Encoder-Decoder framework forms the backbone of the model, with the encoder component utilizing an RGCN to process the protein graphs, and an ESM model to process the protein sequence. A cross-attention mechanism facilitates the exchange of relevant information between the graph-encoded and the sequence-encoded vectors, creating a fused representation synthesizing the structural and textual aspects. The decoder component employs a pre-trained GPT-2 model, to generate detailed and accurate protein descriptions from the fused protein representation. By combining the power of GNNs and transformers, Prot2Text enables a holistic representation of protein function, facilitating the generation of comprehensive protein descriptions.

node $u$ is associated with a feature vector $\boldsymbol{x}_u \in \mathbb{R}^d$, encompassing relevant information such as local structural features, and physicochemical properties of the associated amino-acids. This enables the graph to retain fine-grained information critical to the protein's structure and function.

To model the diverse interactions and relationships between amino acids, we introduce different types of edges connecting the nodes. Therefore, each edge $i = (v, u)$ is associated with an edge type $\boldsymbol{e}_i \in R$. Sequential edges are employed to connect adjacent nodes in the protein sequence, effectively representing the sequential order of amino acids and capturing the linear arrangement of the protein's primary structure. This sequential information is crucial for understanding the folding patterns and functional motifs within the protein. Additionally, we utilize spatial edges to establish connections between nodes that are in close spatial proximity within the 3D structure of the protein. These edges play a pivotal role in encoding the protein's tertiary structure and folding patterns, enabling us to capture the intricate spatial arrangements of amino acids within the protein's core. We further extend the graph construction to include hydrogen bond interactions as an additional edge type. Hydrogen bonds are fundamental non-covalent interactions that are of paramount importance in stabilizing protein structures and enabling specific molecular recognition events. Through the integration of the different edge types, our comprehensive protein graph provides a more holistic and detailed depiction of the protein's structure while capturing both short and long-range interactions.

**Graph Encoding.** To encode the protein graph $G$ into a vector $\boldsymbol{h}_G \in \mathbb{R}^{d_{out}}$, we employ a Relational Graph Convolutional Neural Network(RGCN) (Schlichtkrull *et al.*, 2018), which effectively considers the various edge types present in the graph in the message-passing mechanism. We denote the neighborhood of type $r$ of a vertex $u$ by $\mathcal{N}_r(u)$ such that $\mathcal{N}_r(u) = \{v : (v, u) \in E_r\}$, where $E_r$ is the

set of edges with $r$ edge type. In layer $k$ of the GNN, we update the node representations as follows:

$$\boldsymbol{x}_i^k = \sigma \left( \boldsymbol{W}_{\text{root}}^k \cdot \boldsymbol{x}_i^{k-1} + \sum_{r \in \mathcal{R}} \sum_{j \in \mathcal{N}_r(i)} \frac{1}{|\mathcal{N}_r(i)|} \boldsymbol{W}_r^k \cdot \boldsymbol{x}_j^{k-1} \right),$$ (1)

where $\boldsymbol{W}_{\text{root}}^k$ represents the learnable weight matrix for the root transformation in layer $k$, $\boldsymbol{W}_r^k$ denotes the learnable weight matrix of layer $k$ for relation $r$ and $\sigma(\cdot)$ is an element-wise activation function such as ReLU. This formulation allows nodes to update their representations by incorporating information from neighboring nodes based on the specific edge types, capturing the structural and relational dependencies within the protein graph. To obtain the graph representation from the node representations of the last layer $K$ of the GNN, we apply a mean-pooling layer as follows:

$$\boldsymbol{h}_G = \frac{1}{N} \sum_{i=1}^{N} \boldsymbol{x}_i^K$$ (2)

The resulting vector $\boldsymbol{h}_G$ serves as an informative encoded representation of the protein graph, capturing the essential structural and relational characteristics. This representation plays a crucial role in the subsequent text generation process, where it will be utilized to generate detailed and accurate protein functions.

**Sequence Encoding.** To encode the protein sequence $P_S$, we used ESM2-35M (Lin *et al.*, 2023a) as our base model. ESM2 is a protein language model that uses a transformer-based architecture and an attention mechanism to learn the interaction patterns between pairs of amino acids in the input sequence. This allows the ESM model to capture amino acid sequence evolutionary information about proteins and their properties. In order to achieve uniform representation dimensions for all modalities within the spatial domain, a projection layer is applied after the last hidden layer of the ESM model. This layer functions as a projection layer that transforms the individual amino-acid representations, derived from the ESM embedding dimension, into the graph embedding dimension $d_{out}$. As a result, a matrix denoted as $\boldsymbol{H}_S^0 \in \mathbb{R}^{N, d_{out}}$ is formed, containing the amino-acid representations:

$$\boldsymbol{H}_S^0 = ESM(P_S)\boldsymbol{W}_p$$ (3)

where $\boldsymbol{W}_p$ is a trainable matrix.

**Multimodal Fusion** To obtain the final protein encoding, we utilize a fusion block that combines the representation of each amino acid inside the matrix $\boldsymbol{H}_S^0$ with the graph representation vector $\boldsymbol{h}_G$. The fusion process involves a simple element-wise addition of the two representations, followed by a projection layer. This fusion block enables the integration of information from both the sequence and the graph representations in a straightforward manner. Thus, allowing each amino acid to be contextually enriched with information from the graph representation. Additionally, a normalization layer is applied after each fusion block to maintain stable training and further enhance the learning process. Specifically, for each amino acid representation in $\boldsymbol{H}_S^k$, and the graph representation $\boldsymbol{h}_G$, the fusion block computes the combined representation $\boldsymbol{H}_S^{k+1}$ as follows:

$$\boldsymbol{H}_S^{k+1} = \left( \boldsymbol{H}_S^k + \boldsymbol{1}_n \boldsymbol{h}_G \boldsymbol{W}_V^k \right) \boldsymbol{W}_O^k,$$ (4)

where $\boldsymbol{W}_V^k$, $\boldsymbol{W}_O^k$ are trainable matrices, $\boldsymbol{1}_n$ is a vector of ones, $n$ is the length of the amino-acid sequence, and $k$ is the index to the fusion layer.

By using this fusion block multiple times in the architecture (four times in this case), the model can capture complex interactions and dependencies between the sequence and graph representations, leading to an effective and contextually enriched encoding of the protein data. The fusion block could be seen as a special case of the transformers cross-attention block when the the input from the encoder represents only one token.

**Text Generation** We employed the transformer decoder architecture for generating protein descriptions. We initialized the main components of the decoder, namely the text embedding matrix, self-attention, and language modeling head, with the pre-trained weights of GPT-2. By doing so, we leveraged the GPT-2 model's capacity to grasp the underlying textual semantics. We forward the protein representation obtained from the protein encoder as input to the multi-head cross-attention

module within the transformer decoder. This interaction enabled the model to effectively incorporate context from the protein representation, contributing to the generation of coherent and meaningful protein descriptions. We adopted the identical vocabulary and tokeniser from the GPT-2 model, with the introduction of two unique special tokens. These additional tokens serve as essential markers, enabling the model to discern the precise boundaries of each protein description within the input text. In the training phase, we employed Causal Language Modeling (CLM) as the training objective to optimize our model. Causal Language Modeling involves training the model to predict the next token in a sequence given the preceding tokens. This unidirectional prediction process ensures that the model generates text in a causal manner, without access to future tokens. The maximum length of each description is 256 tokens.

## 4    Experimental Results

**Dataset**    To train the Prot2Text framework using proteins' structures, sequences and textual descriptions, we build a multimodal dataset with $256, 690$ proteins. For each protein, we have three crucial information: the corresponding sequence, the AlphaFold accession ID and the textual description. To build this dataset, we used the SwissProt database (Bairoch and Apweiler, 1996) including the UniProtKB (Consortium, 2016) Release 2022_04. Initially, The SwissProt database in this release has $568, 363$ proteins on which we perform the following: (1) Select the following properties: `name` that gives the full name of the protein, `sequence` that gives the amino-acid sequence of the protein, `AlphaFoldDB` that gives the accession ID of the protein in AlphaFold database, `taxon` and `text` that gives the protein textual description. (2) Eliminate all samples that do not have all three crucial information. (3) Remove all samples with a duplicate amino-acid sequence. (4) Remove all the samples where the textual description contains *"(By Similarity)"*. (5) Apply the CD-HIT clustering algorithm (Li and Godzik, 2006) to create a train/validation/test scheme with $248, 215, 4, 172$ and $4, 023$ proteins respectively. The maximum similarity threshold between the (train, validation test) sets used in the CD-HIT algorithm is 40%. (6) Preprocess the textual description to remove the *"PubMed"* information. The AlphaFoldDB accession is then used to download the protein structure in a ".PDB" file format using version 4 from AlphaFoldDB.

**Baselines.**    In our experimental evaluation, we employed a comprehensive set of baselines to rigorously assess the text generation performance of the Prot2Text framework. Specifically, we compared our approach against unimodal encoders, namely RGCN, ESM, and a vanilla-Transformer trained from scratch. These encoders exclusively focus on either the protein graph or the protein sequence representation. Furthermore, we compared it with a multimodal baseline, RGCN+ESM, that concatenates the graph and sequence representations without fusing the representation of each amino-acid and the structure representation. Finally, we compare with RGCN $\times$ vanilla-Transformer baseline, which has similar architecture as Prot2Text but instead uses a vanilla-Transformer model from scratch instead of the pre-trained ESM2. In all ESM models, we use the last hidden state. The vanilla-Transformer baseline follows the same configuration and number of parameters as the pre-trained ESM2-35M.

**Training Details.**    We implemented all the models using PyTorch and utilized $64$ NVIDIA V100 GPUs for training. We used the AdamW optimizer (Loshchilov and Hutter, 2019) with $\epsilon = 10^{-6}$, $\beta_1 = 0.9$, $\beta_2 = 0.999$, with a learning rate starting from $2.10^{-4}$ and decreasing to zero using a cosine scheduler. We used a warm-up of 6% of the total training steps. We fixed the batch size to four per GPU and we trained the models for 25 epochs. For the GNN encoder, we used 6 layers with a hidden size equal to GPT-2's hidden size (768 for the base model of GPT-2) in each layer. As for the amino-acid sequence toeknisation, We used the same tokeniser and configuration of ESM2 including the hidden layer and hidden dimension. The training for each Base model lasted for approximately 12 hours. All experiments were carried out using the Hugging Face `transformers` library (Wolf *et al.*, 2020).

**Metrics.**    In the experiments, we used several metrics to evaluate the performance of the model in the text generation task. Specifically, we used *BLEU Score* (Papineni *et al.*, 2002) which is a widely used metric for evaluating the quality of machine-generated text. It measures the similarity between the generated text and the reference text based on n-grams. A higher BLEU score indicates better similarity between the generated and reference text. We further used *Rouge-1, Rouge-2* and

| Model | # Params | BLEU Score | Rouge-1 | Rouge-2 | Rouge-L | BERT Score |
|---|---|---|---|---|---|---|
| vanilla-Transformer | 225M | 15.75 | 27.80 | 19.44 | 26.07 | 75.58 |
| ESM2-35M | 225M | 32.11 | 47.46 | 39.18 | 45.31 | 83.21 |
| RGCN | 220M | 21.63 | 36.20 | 28.01 | 34.40 | 78.91 |
| RGCN + ESM2-35M | 255M | 30.39 | 45.75 | 37.38 | 43.63 | 82.51 |
| RGCN × vanilla-Transformer | 283M | 27.97 | 42.43 | 34.91 | 40.72 | 81.12 |
| **Prot2Text$_{BASE}$** | 283M | **35.11** | **50.59** | **42.71** | **48.49** | **84.30** |

Table 1: Test set results for different encoder models, including unimodal encoders such as vanilla-Transformer, ESM2-35M, and RGCN, as well as multimodal encoders such as RGCN×vanilla-Transformer and RGCN+ESM2-35M. All models share the same GPT-2 decoder. Prot2Text$_{BASE}$ achieves the highest performance across all evaluation metrics, including BLEU score, Rouge scores, and BERT Score.

*Rouge-L* scores (Lin, 2004), which measure the overlap of unigrams, bigrams, and longest common subsequence between the generated text and the reference text, respectively. Finally, we used *BERT Score* (Zhang *et al.*, 2020), which measures the similarity between the generated text and the reference text using contextualized word embeddings from a transformer-based model. In our experiments we choose to use BioBERT$_{LARGE}$-cased v1.1 (Lee *et al.*, 2020) to compute the *BERT Score*.

**Results.** We report the results in Table 1, for different encoder models, including unimodal encoders like vanilla-Transformer, ESM2-35M, and RGCN, and multimodal encoders like RGCN × vanilla-Transformer and RGCN + ESM2-35. All models use a GPT-2 decoder. The unimodal vanilla-Transformer baseline, relying solely on the amino-acid sequence of the protein, exhibits the lowest performance across all evaluation metrics. However, we observe a significant improvement in performance when using the unimodal graph encoder RGCN. The RGCN outperforms the vanilla-Transformer by over five absolute points in terms of BLEU score and three points in terms of BERT score. This performance disparity highlights the importance of incorporating structural information through the RGCN encoder for protein's function prediction. On the other hand, leveraging the pre-trained protein language model ESM2-35M instead of initializing the vanilla-Transformer randomly, results in a remarkable improvement in performance. The ESM2-35M encoder leads to a substantial increase of over 16 BLEU score points and 18 Rouge-L points compared to the standard vanilla-Transformer configuration. This notable enhancement can be attributed to the pretraining of ESM2-35M using masked protein modeling, which enables the encoder to capture intricate relationships and patterns within protein sequences. In the context of multimodal protein representation, the evaluation results demonstrate that Prot2Text$_{BASE}$ exhibits superior performance across all assessment metrics. Notably, it achieves the highest BLEU Score of $35.11$, the highest Rouge-1 score of $50.59$, the highest Rouge-2 score of $42.71$, the highest Rouge-L score of $48.49$, and the highest BERT Score of $84.3$. These outcomes highlight the effectiveness of fusing protein structure and amino-acid information in a multimodal manner. The incorporation of protein structure, facilitated by the Relational Graph Convolutional Network (RGCN) with the sequential representations of amino-acids from ESM2-35, significantly enhances the overall performance across all evaluation metrics. This improvement is attributed to the enriched understanding of proteins achieved through the synergy of these two modalities. Furthermore, the efficacy of the multimodal fusion approach is corroborated by the results obtained from RGCN × vanilla-Transformer. Introducing structural information using RGCN to the randomly initialized vanilla-Transformer yields a substantial improvement of over 10 BLEU score points compared to using the vanilla-Transformer alone, and more than 6 BLEU score points improvement over using RGCN in isolation. Finally, to show the importance of the fusion block in the Prot2Text framework, we compare it against RGCN + ESM2-25, which concatenates the protein structure representation to the amino-acids representation. In this case, the graph representation will simply be passed to the decoder alongside the ESM output. We notice that using this strategy leads to slightly worse results than using the ESM alone. This not only provides backing for the selection of the fusion block employed in Prot2Text, but also suggests that indiscriminately increasing the overall parameter count of the model could potentially lead to a degradation in its performance.

**Ablation Study: Scaling to Larger Models.** We conducted an ablation study to assess the performance of our Prot2Text framework as we varied the number of parameters. The primary objective of this experiment was to evaluate the benefits of employing larger models in terms of generating more

| Model | # Params | BLEU Score | Rouge-1 | Rouge-2 | Rouge-L | BERT Score | Inference Time |
|---|---|---|---|---|---|---|---|
| **Prot2Text**$_{\text{SMALL}}$ | 256M | 30.01 | 45.78 | 38.08 | 43.97 | 82.60 | 1,225 |
| **Prot2Text**$_{\text{BASE}}$ | 283M | 35.11 | 50.59 | 42.71 | 48.49 | 84.30 | 1,379 |
| **Prot2Text**$_{\text{MEDIUM}}$ | 398M | **36.51** | 52.13 | 44.17 | 50.04 | 84.83 | 1,334 |
| **Prot2Text**$_{\text{LARGE}}$ | 898M | 36.29 | **53.68** | **45.60** | **51.40** | **85.20** | 1,667 |

Table 2: Test set results for different size variations of Prot2Text. Larger models outperform their smaller counterparts across most evaluation metrics, indicating the benefits of employing larger language models in the Prot2Text framework. The Prot2Text$_{MEDIUM}$ model, strikes an optimal balance between performance and computational efficiency. This configuration demonstrates improved performance compared to the smaller model while still maintaining reasonable computational costs. The inference time is in seconds for text generation of each model on the whole test set. The inference time here is computed during text generation using two NVIDIA RTX 6000 with 48GB memory in parallel and batch size of four per device.

accurate and detailed textual representations of protein's function. To conduct the ablation study, we systematically varied the size of the protein language model (ESM). Where Prot2Text$_{SMALL}$, Prot2Text$_{BASE}$, Prot2Text$_{MEDIUM}$ and Prot2Text$_{LARGE}$ use ESM2-8M, ESM2-35M, ESM2-150M and ESM2-650M respectively. We evaluated each configuration on the same test set of proteins and used the same evaluation metrics as described earlier. The results of the ablation study, presented in Table 2, show a trend of performance improvement as we scale up the model's architecture. Larger versions of ESM outperformed their smaller counterparts in most evaluation metrics. The increase in model size led to more accurate and relevant descriptions, indicating the benefit of leveraging larger language models in the Prot2Text framework. Yet, complementary analysis including corresponding computation time showed an increase in the inference cost following the use of larger models (higher number of parameters). Therefore, **Prot2Text**$_{MEDIUM}$ ($398M$ parameters) is a good trade-off striking the balance between performance and time cost.

## 5    Conclusion

In conclusion, our paper introduces Prot2Text, a pioneering multimodal framework, for the accurate prediction of a protein's function in free text format, from graph and sequential input. By reformulating the task as free-text prediction, we address the limitations of traditional classification-based methods, allowing for a more nuanced and in-depth understanding of a protein's functionality. Leveraging the power of GNNs and pretrained language models, we integrate structural and textual protein information, resulting in highly detailed and coherent generated protein descriptions. The release of a comprehensive multimodal protein dataset further empowers the scientific community to benchmark and advance the field of protein function prediction in free text format. This innovative approach opens new horizons for research and applications in drug discovery, protein engineering, and various biological sciences, with the potential to revolutionize our understanding of proteins' functions.

## 6    Limitations and Future Work

One limitation of our proposed Prot2Text model is that the RGCN encoder is not pretrained. Unlike the ESM encoder, which benefits from pretraining on a large corpus, the RGCN encoder lacks this initial knowledge. As a result, the RGCN encoder might struggle to capture complex patterns and may not fully leverage the underlying protein data, potentially leading to suboptimal performance. To address this limitation, we aim to explore pretraining techniques specifically tailored for graph neural networks. This could involve pretraining the RGCN encoder on auxiliary graph-related tasks, leveraging graph-level or node-level information to build a foundational understanding of protein structures.

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

# A    Training Details

## A.1    Tokenization

**Proteins Textual Description**    The training dataset consists of 256,690 proteins with unique amino-acid sequence. However, some proteins have the same description. In total, the training dataset has 48,251 unique function descriptions. The average number of tokens per description is 57.51. We chose to truncate all the descriptions during the tokenization to a maximum length of 256 since this number of tokens covers 98.7% of all the descriptions as we can see in figure 2.

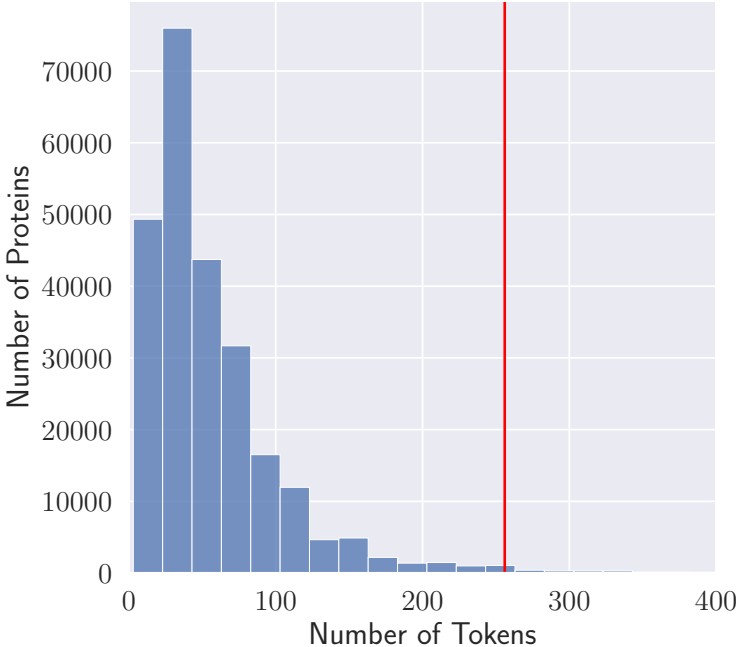

Figure 2: Analyzing Protein Description Lengths: Distribution of Tokens per Sample with Threshold Highlight at 256 tokens (in red)

**Tokenizer**    The Prot2Text tokenizer is an instance of the GPT2 tokenizer with two additional tokens. In GPT2 model, the pad token, the start of sequence token and the end of sequence token share the same index. As the Prot2Text architecture is an encoder-decoder architecture, we chose to separate the three tokens by adding two extra tokens representing the start of sequence and the end of sequence. For both added tokens, we equally need to add the corresponding embedding to the GPT2 word embedding matrix while keeping the rest of the matrix intact.

## A.2    Text Generation

To generate the protein textual description during and after the training, we used the generation function implemented in the transformers library. We used the default generation parameters of `length_penalty=1.0`, `no_repeat_ngram_size=0` and `early_stopping=False`. The text generation was done during the training on the validation set each 500 training steps using greedy search (number of beams equal to one) with maximum length of 256 tokens per sample. However, different configuration could be used leading to multiple functions. For example, the generated text in figure 3 from the main paper uses `length_penalty=2.0`, `no_repeat_ngram_size=3` and `early_stopping=True` using Prot2Text$_{BASE}$.

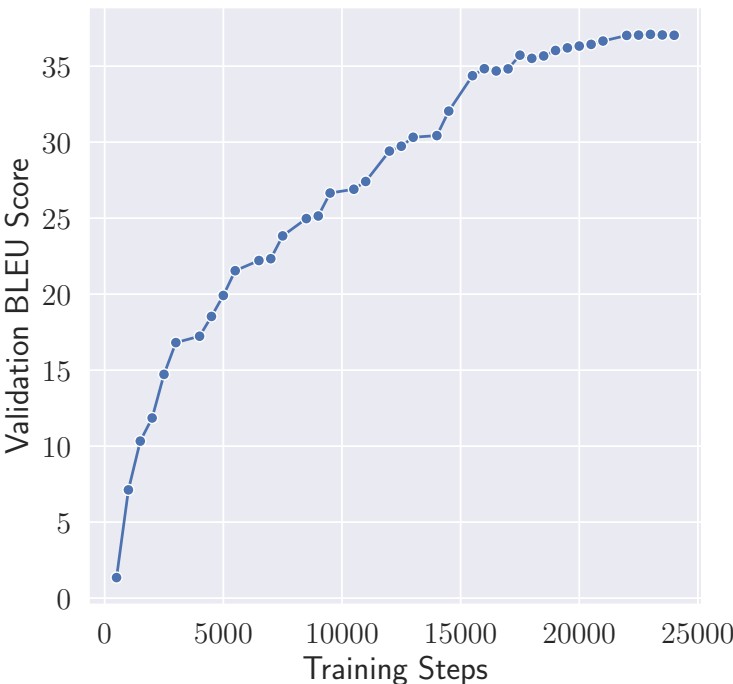

Figure 3: Tracking Prot2Text$_{BASE}$ BLEU Score Progression on Validation Set Across Training Iterations. Higher is better.

Figure 3 shows the BLEU score validation throughout the training for the Prot2Text$_{BASE}$ model. The validation BLEU score start to stabalize after the 20th epochs reaching the best validation BLEU score of 37.09 at the step 23000.

### A.3   CO2 Emission Related to Experiments

Experiments were conducted using a private infrastructure, which has a carbon efficiency of 0.057 kgCO$_2$eq/kWh. A cumulative of 23000 hours of computation was performed on hardware of type Tesla V100-SXM2-32GB (TDP of 300W). Total emissions are estimated to be 393.3 kgCO$_2$eq of which 0 percents were directly offset. Estimations were conducted using the MachineLearning Impact calculator[1]

## B   Prot2Text Performance with respect to sequence alignment

In Figure 4 we report the performance of all Prot2text models with respect to different similarity thresholds. Where the similarity represents the highest alignment score between the amino-acid sequences of the test and train sets using BLAST identity. We observe that for test proteins with low similarity scores with the train set (between 20% and 30%) and for proteins with no counterpart in the train set, the Prot2Text$_{MEDIUM}$ is the dominant one while for higher similarity scores Prot2Text$_{LARGE}$ performs better.

## C   Visualization of Generated Descriptions.

To gain deeper insights into the quality of the generated proteins' functions by our *Prot2Text* framework, we provide in Figure 5 a textual comparison of the pre-defined labels and generated text outputs for a selected set of proteins from the test set. It illustrates a comparison between the ground truth and the corresponding descriptions generated by *Prot2Text* for three different proteins (*P36108*, *Q8NG08*

---

[1]https://mlco2.github.io/impact#compute

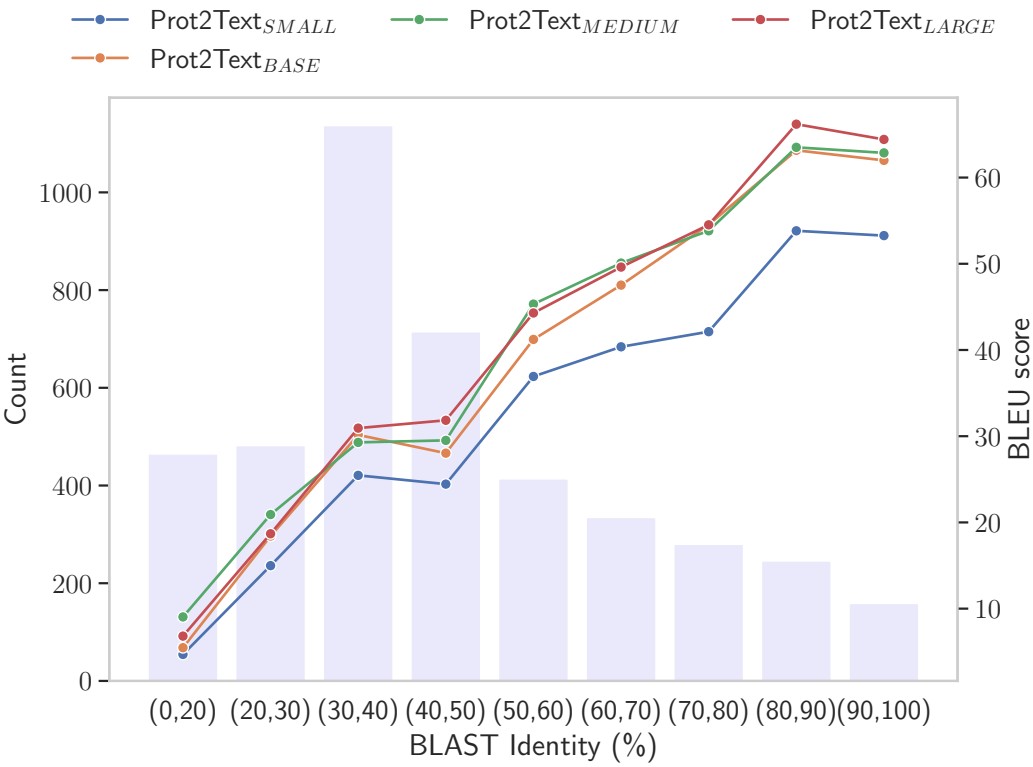

Figure 4: The test BLEU score for Prot2Text models as a function of the percentage identity using BLAST hit between the test and the train sets.

and *P35713*) along with each protein's name, amino acid sequence and 3D structural representation. The results indicate a successful detailed reconstruction of the different proteins' functions including richer information than the known description. Following, Fig. 5 showcases the model's ability to generate coherent and informative free-text descriptions that align closely with the ground truth annotations.

| Protein | Labeled Text | Generated Text |
|---|---|---|
| 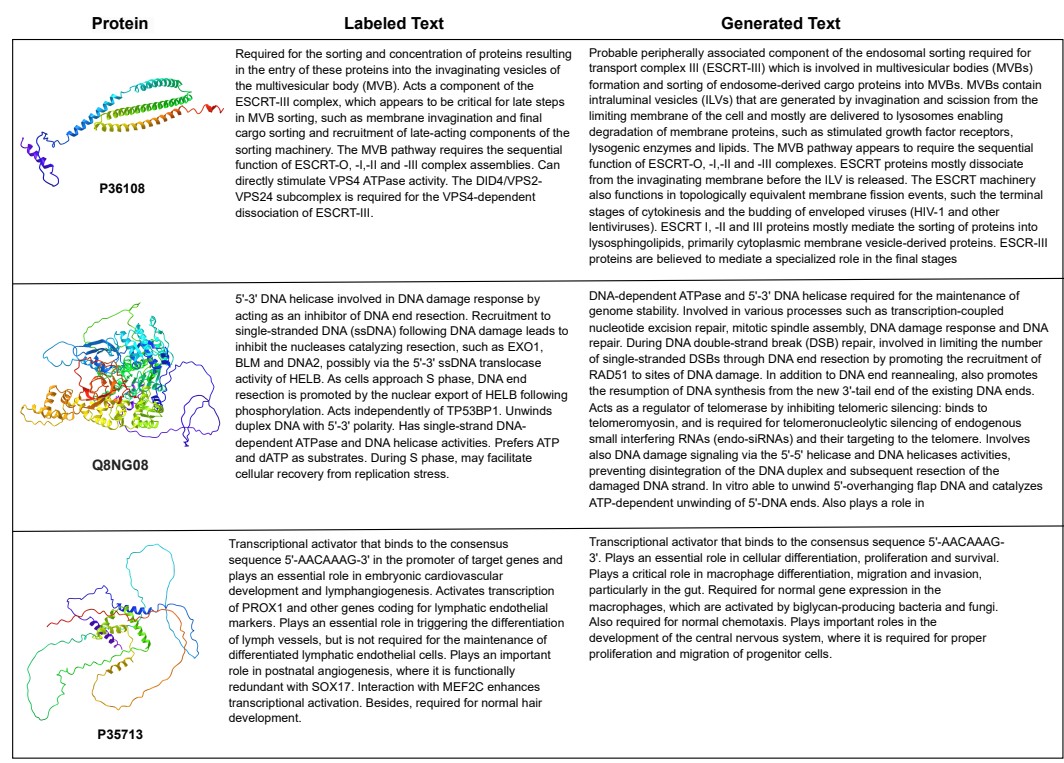 **P36108** | Required for the sorting and concentration of proteins resulting in the entry of these proteins into the invaginating vesicles of the multivesicular body (MVB). Acts a component of the ESCRT-III complex, which appears to be critical for late steps in MVB sorting, such as membrane invagination and final cargo sorting and recruitment of late-acting components of the sorting machinery. The MVB pathway requires the sequential function of ESCRT-O, -I,-II and -III complex assemblies. Can directly stimulate VPS4 ATPase activity. The DID4/VPS2-VPS24 subcomplex is required for the VPS4-dependent dissociation of ESCRT-III. | Probable peripherally associated component of the endosomal sorting required for transport complex III (ESCRT-III) which is involved in multivesicular bodies (MVBs) formation and sorting of endosome-derived cargo proteins into MVBs. MVBs contain intraluminal vesicles (ILVs) that are generated by invagination and scission from the limiting membrane of the cell and mostly are delivered to lysosomes enabling degradation of membrane proteins, such as stimulated growth factor receptors, lysogenic enzymes and lipids. The MVB pathway appears to require the sequential function of ESCRT-O, -I,-II and -III complexes. ESCRT proteins mostly dissociate from the invaginating membrane before the ILV is released. The ESCRT machinery also functions in topologically equivalent membrane fission events, such the terminal stages of cytokinesis and the budding of enveloped viruses (HIV-1 and other lentiviruses). ESCRT I, -II and III proteins mostly mediate the sorting of proteins into lysosphingolipids, primarily cytoplasmic membrane vesicle-derived proteins. ESCR-III proteins are believed to mediate a specialized role in the final stages |
| **Q8NG08** | 5'-3' DNA helicase involved in DNA damage response by acting as an inhibitor of DNA end resection. Recruitment to single-stranded DNA (ssDNA) following DNA damage leads to inhibit the nucleases catalyzing resection, such as EXO1, BLM and DNA2, possibly via the 5'-3' ssDNA translocase activity of HELB. As cells approach S phase, DNA end resection is promoted by the nuclear export of HELB following phosphorylation. Acts independently of TP53BP1. Unwinds duplex DNA with 5'-3' polarity. Has single-strand DNA-dependent ATPase and DNA helicase activities. Prefers ATP and dATP as substrates. During S phase, may facilitate cellular recovery from replication stress. | DNA-dependent ATPase and 5'-3' DNA helicase required for the maintenance of genome stability. Involved in various processes such as transcription-coupled nucleotide excision repair, mitotic spindle assembly, DNA damage response and DNA repair. During DNA double-strand break (DSB) repair, involved in limiting the number of single-stranded DSBs through DNA end resection by promoting the recruitment of RAD51 to sites of DNA damage. In addition to DNA end reannealing, also promotes the resumption of DNA synthesis from the new 3'-tail end of the existing DNA ends. Acts as a regulator of telomerase by inhibiting telomeric silencing: binds to telomeromyosin, and is required for telomeronucleolytic silencing of endogenous small interfering RNAs (endo-siRNAs) and their targeting to the telomere. Involves also DNA damage signaling via the 5'-5' helicase and DNA helicases activities, preventing disintegration of the DNA duplex and subsequent resection of the damaged DNA strand. In vitro able to unwind 5'-overhanging flap DNA and catalyzes ATP-dependent unwinding of 5'-DNA ends. Also plays a role in |
| **P35713** | Transcriptional activator that binds to the consensus sequence 5'-AACAAAG-3' in the promoter of target genes and plays an essential role in embryonic cardiovascular development and lymphangiogenesis. Activates transcription of PROX1 and other genes coding for lymphatic endothelial markers. Plays an essential role in triggering the differentiation of lymph vessels, but is not required for the maintenance of differentiated lymphatic endothelial cells. Plays an important role in postnatal angiogenesis, where it is functionally redundant with SOX17. Interaction with MEF2C enhances transcriptional activation. Besides, required for normal hair development. | Transcriptional activator that binds to the consensus sequence 5'-AACAAAG-3'. Plays an essential role in cellular differentiation, proliferation and survival. Plays a critical role in macrophage differentiation, migration and invasion, particularly in the gut. Required for normal gene expression in the macrophages, which are activated by biglycan-producing bacteria and fungi. Also required for normal chemotaxis. Plays important roles in the development of the central nervous system, where it is required for proper proliferation and migration of progenitor cells. |

Figure 5: **Ground-truth labels vs text-free Generated functions**: A textual comparison of the pre-defined labels and generated text outputs for 3 different proteins from the test set.

