# OpenReview forum: "Prot2Text: Multimodal Protein's Function Generation with GNNs and Transformers"
_NeurIPS.cc/2023/Workshop/AI4Science — NeurIPS2023-AI4Science Poster_

### Official Review · Reviewer_HeZm · 2023-10-08

**Rating:** 6
**Confidence:** 4

**Review:**

This paper introduces an innovative approach to protein function prediction. The paper leverages a combination of GNNs and Large Language Models (LLMs) within an encoder-decoder framework to integrate diverse data types, including protein sequence, structure, and textual annotations.

Strengths:

Methodology is novel: The paper effectively combines GNNs and LLMs, harnessing the power of both approaches to create a more comprehensive representation of proteins' functions.
Empirical Evaluation: The evaluation is comprehensive. The authors provide empirical evidence of the model's effectiveness using a real-world dataset, reinforcing the practical utility of Prot2Text.


Minor point: While the exact definition of LLMs varies, it is generally acknowledged that GPT2 is NOT considered an LLM (Line 44-45).

---

### Official Review · Reviewer_p29C · 2023-10-25
**Concern about the evaluation**

**Rating:** 6
**Confidence:** 3

**Review:**

The paper titled "Prot2Text: Multimodal Protein's Function Generation with GNNs and Transformers" presents an innovative approach to predict a protein's function using a multimodal framework integrating Graph Neural Networks (GNNs) and Large Language Models (LLMs). While the research introduces a promising direction in protein function prediction, there are concerns regarding the evaluation methodology and the practical implications of generating textual descriptions for proteins.

1. Multimodal Fusion Method (Line 182): The paper mentions employing an element-wise addition for multimodal fusion. While this might be a straightforward approach, I am unfamiliar with its widespread use in current literature. It would be beneficial if the authors can provide references or justifications for choosing this particular strategy over others. Is element-wise addition a commonly adopted technique in the field?
2. Rationale Behind Textual Descriptions: The paper suggests the use of free text for describing protein functions. While this might offer a richer description, the practical advantages need clarification. Given that the primary target audience for such predictions would be biology experts, how does a textual description aid them compared to the more concise categorical classifications? Is there a significant benefit in terms of information richness or interpretability?
3. Evaluation Concerns:
- The choice of BLEU score as the evaluation metric raises some doubts. While BLEU is a standard metric in machine translation tasks, its appropriateness in this context is questionable. The essence of a protein's function might be captured in a few words, with the rest of the textual description potentially being superfluous or redundant. Would it not be more beneficial to evaluate based on embeddings, which might capture semantic relationships better?
- Furthermore, the paper lacks a comparison between the proposed method and the conventional multi-class accuracy in predicting protein function. Such a comparison would provide insights into the actual benefits of the proposed approach. Is there a significant improvement in accuracy, or are there other advantages, such as interpretability or richness of information, which might justify the shift to a text-based description?